# Multiple spatial reference frames underpin perceptual recalibration to audio-visual discrepancies

**David Mark Watson**[1,2]*, **Michael A. Akeroyd**[3], **Neil W. Roach**[1], **Ben S. Webb**[1]

**1** School of Psychology, University of Nottingham, Nottingham, United Kingdom, **2** Department of Psychology, University of York, York, United Kingdom, **3** Hearing Sciences, Division of Clinical Neuroscience, School of Medicine, University of Nottingham, Nottingham, United Kingdom

* david.watson@york.ac.uk

**Data Availability Statement:** All data, code and materials for both main experiments and the pilot experiment can be accessed on the Open Science Framework: https://osf.io/7gdtp/ (DOI: 10.17605/OSF.IO/7GDTP).

## Abstract

In dynamic multisensory environments, the perceptual system corrects for discrepancies arising between modalities. For instance, in the ventriloquism aftereffect (VAE), spatial disparities introduced between visual and auditory stimuli lead to a perceptual recalibration of auditory space. Previous research has shown that the VAE is underpinned by multiple recalibration mechanisms tuned to different timescales, however it remains unclear whether these mechanisms use common or distinct spatial reference frames. Here we asked whether the VAE operates in eye- or head-centred reference frames across a range of adaptation timescales, from a few seconds to a few minutes. We developed a novel paradigm for selectively manipulating the contribution of eye- versus head-centred visual signals to the VAE by manipulating auditory locations relative to either the head orientation or the point of fixation. Consistent with previous research, we found both eye- and head-centred frames contributed to the VAE across all timescales. However, we found no evidence for an interaction between spatial reference frames and adaptation duration. Our results indicate that the VAE is underpinned by multiple spatial reference frames that are similarly leveraged by the underlying time-sensitive mechanisms.

## Introduction

In dynamic multisensory environments, the human perceptual system integrates sensory information across multiple modalities whilst also correcting for sensory discrepancies between those modalities [1]. Such discrepancies can lead to a perceptual recalibration of the sensory environment. For instance, exposure to temporally offset visual, auditory, and/or tactile stimuli can bias the perception of timing amongst those stimuli so as to reduce the perceived asynchrony [2–4]. Similarly, spatial discrepancies between audio-visual stimuli induce a spatial recalibration such that the perception of auditory locations is biased in the direction of the visual offset [5–9]: the "ventriloquism aftereffect" (VAE). The VAE is observed following a diverse range of timescales of adaptation, from several minutes [4, 7, 10, 11] down to just a few seconds or even a single stimulus presentation [12–14]. In a recent study we demonstrated

**Funding:** This work was funded by a Leverhulme Trust grant (RPG-2016-077; https://www.leverhulme.ac.uk/) awarded to N.W.R and B.S.W. M.A.A. is supported by a Medical Research Council grant (MR/S002898/1; https://mrc.ukri.org/). The funders had no role in study design, data collection and analysis, decision to publish, or preparation of the manuscript.

**Competing interests:** The authors have declared that no competing interests exist.

that the VAE is underpinned by multiple recalibration mechanisms operating over different timescales, such that multiple VAEs at different temporal scales may be maintained simultaneously [15]. Nevertheless, it remains unclear whether longer- versus shorter-timescale sensitive mechanisms rely on distinct or shared spatial reference frames.

The VAE relies on integrating visual and auditory spatial information, yet these two modalities originate in very different spatial reference frames. Auditory inputs are derived from, and are primarily encoded within, a head-centred reference space. By contrast, visual signals are derived from an eye-centred reference space. Visual signals remain encoded in retinotopic reference frames in early visual cortex [16] (although some saccade-dependent signals are observed even in primary visual cortex [17]), whilst both retinotopic and spatiotopic reference frames are observed in later processing stages—particularly in dorsal parietal regions [18, 19]. Furthermore, multisensory parietal regions are also implicated in the concurrent representation of visual and auditory spatial receptive fields [20]. Audio-visual spatial recalibration effects themselves are associated with responses in primary auditory cortices [21–23], but are also mediated by higher-level multisensory regions including those in parietal cortex [24].

Thus, in eliciting the VAE, head-centred auditory signals could conceivably be combined with visual signals that are represented in either eye- or head-centred reference frames (or both). This issue was investigated in a study by Kopčo and colleagues [25], who adapted participants to spatially disparate audio-visual stimuli presented within a central range of azimuths, whilst maintaining constant fixation at an off-centre location. During test trials, participants either continued fixating at the adapting position, or shifted fixation to the opposite hemifield. Maintaining fixation at the adapting location yielded a spatial tuning curve of the VAE, such that effects were largest around the adapted region of space and decreased outside it. When shifting fixation position, the tuning curve partially (though not totally) shifted with the direction of the saccade, suggesting a combined influence of eye- and head-centred visual reference frames on the VAE. This study also provided preliminary evidence of a shift between reference frames over time: the VAE initially appeared mostly head-centred, but progressed towards using combined eye- and head-centred frames following more sustained adaptation. Nevertheless, this particular finding resulted from a supplementary analysis, and the study design was not optimised for assessing this hypothesis as it did not explicitly include different durations of adaptation. Thus, the temporal dynamics of eye- versus head-centred contributions to the VAE remain unclear. Furthermore, a recent replication attempt of this experiment from the same group found little evidence for eye-centred contributions [26].

Here, we report two experiments aimed at extending the findings of [25] by explicitly testing for a shift in the contribution of eye- versus head-centred visual reference frames following different durations of adaptation, ranging from a few seconds up to a few minutes. In the first experiment, we present a novel paradigm for selectively testing the contributions of eye- and head-centred visual signals by manipulating the location of auditory stimuli relative to either the participants' head orientation or fixation position. During adaptation, participants maintained their fixation on moving visual targets whilst their heads remained stationary, such that the visual location was variable in head-centred co-ordinates but remained fixed at the same foveal position in eye-centred co-ordinates. When the auditory stimulus is positioned relative to the fixated visual stimulus, combining head-centred auditory signals with head-centred visual signals will yield a consistent audio-visual spatial disparity, whilst combination with eye-centred visual signals will produce inconsistent disparities. When the auditory stimulus is instead located relative to the head orientation, the reverse is true. In this way, we were able to selectively manipulate the consistency of audio-visual spatial disparities within a specific reference frame. Assuming that eliciting a robust VAE depends on audio-visual spatial disparities being largely consistent, this design allows us to selectively maintain or disrupt the

contribution of each visual reference frame to sensory recalibrations. By testing the VAE following different durations of adaptation under each of these conditions, we could determine the relative contributions of eye- and head-centred visual reference frames to the VAE over different timescales. If recalibration mechanisms tuned to different timescales entail a shift between spatial frames, we would expect an interaction between the fixation condition and adaptation duration. Finally, in a second experiment we quantified the extent to which adaptation to consistent versus inconsistent audio-visual disparities drive the VAE.

## Experiment 1: Reference frames of spatial recalibration

### Methods

**Participants.** Twenty-three participants took part in the study. Two participants were excluded due to difficulties in localising the auditory stimuli, and a further participant was excluded due to a hardware malfunction in one experimental session, leaving a total sample of 20 participants (9 male, 11 female, median age = 27, age range = 23 – 42). The study was approved by the ethics committee of the School of Psychology at the University of Nottingham (ethics approval number: 902) and conducted in accordance with the guidelines and regulations of this committee and the Declaration of Helsinki. Participants provided informed written consent before participating in the study. Participants received an inconvenience allowance of £10/hour to compensate their time.

The sample size was determined by an *a priori* power analysis. An estimated effect size was obtained from an earlier pilot study of 5 participants conducted prior to commencing data collection for the current experiment. This pilot employed the same design as the current experiment (see below), except that only two adaptation durations were included (35s and 140s). Power analyses were based on the fixation-condition by adaptation-duration interaction in an ANOVA of the VAE magnitude data, as this is the primary effect of interest with regards to our hypothesis. This interaction ($F(1.59, 6.36) = 3.32$, $p = .108$) yielded an effect size of Cohen's $f = 0.91$. This effect size was entered into a power analysis using G*Power (v3.1; [27]), accounting for the 3 fixation-condition and 4 adaptation-duration levels to be employed in the main experiment. This revealed a minimum sample size of 18 subjects would be required to achieve 80% power with an alpha criterion of 0.05. We thus aimed for a slightly larger sample size of 20. Note that an update to the first-level regression modelling approach (see *Methods*: *Deviations from pre-registration* section) slightly reduces the interaction effect size from the one reported above and in our pre-registration; updated power calculations are provided in S1 Methods.

**Materials.** The stimuli and experimental apparatus follow those described in our previous work [15]. Visual stimuli were projected onto a curved screen (radius = 2.5 m, height = 2 m ≈ 44˚ elevation) wrapping 180˚ in azimuth around the participant. Three interleaved projectors projected video feeds onto the screen, and Immersaview's Sol7 software (https://www. immersaview.com/) blended the feeds and corrected for the curvature of the screen. Visual stimuli during adaptation phases comprised 2-dimensional luminance Gaussian blobs (FWHM = 5˚ / σ = 2.12˚ of visual angle) presented at 0˚ elevation and across a range of azimuths. Stimuli were presented for a duration of 500 ms during which they were sinusoidally contrast modulated (rate = 6 Hz, between 50% and 100% of maximum contrast). During test phases a visual marker used for making responses was presented, which subtended 1˚ of visual angle and the full height of the screen. Throughout the whole experiment, a pair of vertical fixation lines were presented above and below 0˚ elevation so as not to occlude the Gaussian blobs. Fixation lines were either presented at 0˚ azimuth (eye+head-consistent condition) or at the current location of the Gaussian blob (eye- and head-consistent conditions). The colour of

the lines indicated the current phase of the experiment: displayed as red and blue during adaptation and test phases respectively.

Audio stimuli comprised 500ms pink-noise bursts (100–4000 Hz bandpass) which were sinusoidally amplitude modulated at a rate of 6 Hz and with a depth of 3 dB. Stimuli were sampled at 44.1 kHz and presented binaurally over Sennheiser HD265 headphones (average listening level = 62 dB(A) SPL at 0˚ azimuth). Auditory azimuths were emulated via head-related transfer functions (HRTFs) derived from the MIT Kemar database [28], providing azimuths between ±90˚ in 5˚ intervals. To encourage perceptual binding of visual and auditory stimuli, the image-source method [29] was used to add virtual reverberations to the auditory signals to simulate sound sources at the distance of the screen. Following the dimensions of the testing environment, the participant was modelled as sitting 1.5 m from the back and in the horizontal centre of a 4.2 x 5.2 m room. Sources were simulated from a 2.5 m radius arc in front of the participant, equivalent to the distance of the projection screen. Reverberations assumed walls with a uniform absorbance of 0.2, yielding up to 5 reflections. An impulse response function was constructed by collecting the incoming pulses at the participant's location, which was then in turn convolved with each of the MIT Kemar HRTFs. This produced a new set of HRTFs that could then be convolved with a given auditory signal to simulate both the azimuth and distance of the source given the sound reverberations. These HRTFs provided an effective simulation of auditory location, as indicated by the strong linear relationship between stimulus azimuths and participants' perception of those azimuths (S1–S4 Figs). The auditory signals were first gated by 25ms raised-cosine ramps at onsets and offsets before being convolved with the HRTFs.

**Design and procedure.** The experimental procedures expand upon those described in our previous work [15]. Participants were instructed to orient their head directly forward and to maintain their fixation position between the vertical fixation lines throughout the full experiment. A chin-rest was used to assist participants in maintaining their head position. A fully within-subjects design was employed comprising three main factors: fixation-condition (eye +head-consistent, eye-consistent, and head-consistent), adaptation-duration (35, 70, 105, and 140 s), and audio-visual spatial disparity (±20˚).

During adaptation phases, both the fixation and stimulus positions varied according to the three fixation conditions, each designed to selectively manipulate the consistency of the audio-visual spatial disparities produced by combination of head-centred auditory signals with eye- and head-centred visual signals (Fig 1). Note that head position remained fixed dead-ahead in all conditions. In the eye+head-consistent condition, participants maintained central fixation throughout, whilst auditory stimuli were spatially offset relative to the location of the current visual stimulus. Thus, a consistent audio-visual spatial disparity would be obtained using both eye- and head-centred visual signals. In both the eye- and head-consistent conditions, participants instead maintained fixation at the current location of the visual stimulus, such that the visual stimulus now always occurred at the same foveal eye-centred location, but still varied in terms of head-centred locations. In the eye-consistent condition, the auditory stimulus was repeatedly presented at a location relative to 0˚ azimuth (±20˚ spatial offset). The head-centred auditory position thus remained spatially consistent with the eye-centred visual position (both appeared relative to centre), but inconsistent with the head-centred visual position (visual location varied whilst the auditory location did not). Conversely, in the head-consistent condition, the auditory stimulus was presented relative to the visual stimulus location (as per the eye +head-consistent condition). The head-centred auditory position was thus spatially consistent with the head-centred visual position (auditory and visual locations varied around the head together), but inconsistent with the eye-centred visual position (visual location remained central throughout whilst the auditory location did not). A message presented on screen at the

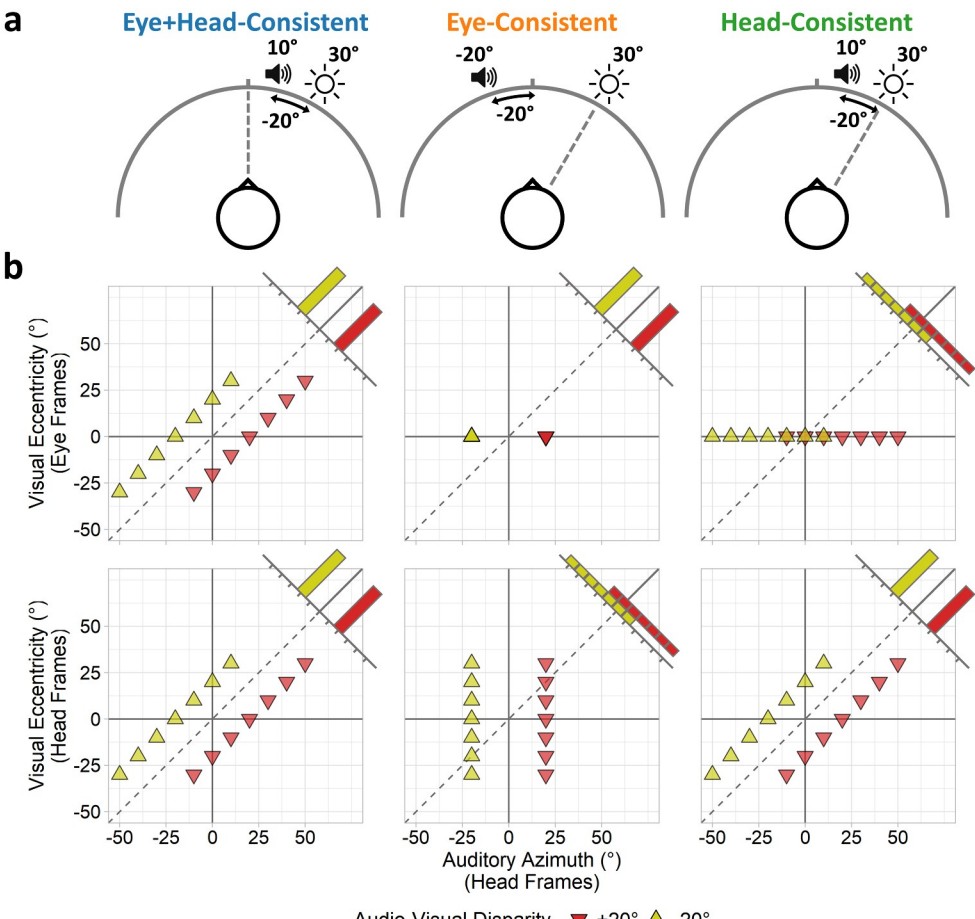

**Fig 1. Experiment 1: Illustration of conditions.** In the eye+head-consistent condition (left column), participants maintain central fixation and auditory stimuli are positioned relative to the visual stimulus location: consistent audio-visual spatial disparities are obtained from both eye- and head-centred visual frames. In the eye-consistent condition (middle column), participants maintain fixation at the visual stimulus location, and auditory stimuli are located relative to 0˚ azimuth: audio-visual spatial disparities remain consistent using eye-centred visual frames, but appear inconsistent using head-centred visual frames. In the head-consistent condition (right column), participants fixate the visual stimulus, and auditory stimuli are positioned relative to the visual stimulus: audio-visual spatial disparities remain consistent using head-centred visual frames, but appear inconsistent using eye-centred visual frames. (a) Stimuli and fixation positions for an example trial pairing a visual stimulus at +30˚ eccentricity with an auditory stimulus offset -20˚ leftward. (b) Stimulus locations: head-centred auditory azimuth plotted against visual eccentricity represented in eye-centred (top-row) and head-centred frames (bottom-row). Spatially consistent stimuli should lie parallel to the dotted line. Corner histograms illustrate distributions of audio-visual disparities.

start of each block informed participants whether the fixation would remain central or move with the visual stimulus in that block.

During each adaptation phase, visual and auditory stimuli were presented synchronously for a duration of 500ms with a 300ms inter-stimulus interval. Visual and audio stimuli were sinusoidally modulated together to promote perceptual binding between them. Audio-visual pairs were presented 5 times consecutively at each azimuth to assist participants in allocating spatial attention [7]. A 1 second ISI was included between each set of 5 presentations to allow participants sufficient time to move their fixation position (as required) before presentations at the next azimuth commenced. Audio-visual pairs were presented with either a -20˚ (leftward audio shift) or +20˚ (rightward audio shift) offset in azimuth. Audio spatial offsets were

**Table 1. Experiment 1: Summary of head, fixation, and stimulus positions for each adaptation condition.**

| Condition | Head position | Fixation position | Visual stimuli positions | Auditory stimuli positions |
|---|---|---|---|---|
| Eye+Head-Consistent | 0˚ absolute | 0˚ absolute | 0˚, ±10˚, ±20˚, ±30˚ absolute | ±20˚ of visual stimulus |
| Eye-Consistent | 0˚ absolute | Visual stimulus | 0˚, ±10˚, ±20˚, ±30˚ absolute | ±20˚ absolute |
| Head-Consistent | 0˚ absolute | Visual stimulus | 0˚, ±10˚, ±20˚, ±30˚ absolute | ±20˚ of visual stimulus |

applied relative to either the visual stimulus location (eye+head- and head-consistent conditions) or to 0˚ azimuth (eye-consistent condition). For example, under the eye+head- and head-consistent conditions, an audio-visual pair presented at +30˚ azimuth with a -20˚ auditory offset would comprise a visual stimulus at +30˚ and an audio stimulus at +10˚ azimuth. The same pair in the eye-consistent condition would comprise a visual stimulus at +30˚ and an auditory stimulus at -20˚ azimuth. Visual stimuli locations ranged between -30˚ (left) and +30˚ (right) eccentricity in 10˚ increments (7 locations total). A summary of the head, fixation, and stimuli positions during adaptation phases is presented in Table 1. Each adaptation phase comprised 1, 2, 3, or 4 passes over all locations (corresponding to 35, 70, 105, or 140 s) according to the adaptation-duration condition of the block. The order of locations was randomised within each pass.

During each test phase, audio stimuli (specifications as per adaptation phase) were presented unimodally. The vertical fixation lines were presented centrally (0˚ azimuth), and participants maintained central fixation, throughout all test phases. On each trial a random stimulus location was selected from a uniform distribution between ±30˚ azimuth in 5˚ steps (13 azimuths total). Following each stimulus presentation, participants reproduced their perception of the auditory azimuth by moving an on-screen visual marker left and right via a trackball mouse and entering their response via mouse click. An inter-trial interval of 200ms was included between each response and subsequent stimulus presentation.

Conditions were presented within a blocked-design, with each of the 24 conditions (3 fixation-conditions × 4 adaptation-durations × 2 spatial-disparities) allocated to one block of the experiment. The block order was fully randomised for each participant independently. Each block included 4 cycles of alternating adaptation and test phases, with each test phase comprising 10 trials. Each participant thus provided 40 responses per condition/block. Following each test phase, a 10 second countdown was presented on screen before the next adaptation phase to mitigate adaptation effects carrying over between cycles. A further break period of at least one minute was enforced between each block. Participants were able to split the experiment over multiple testing sessions at their convenience; participants typically completed the experiment across 5 sessions each comprising 4 to 5 blocks. Average block durations were 3m58s, 6m15s, 8m36s, and 10m57s for the 35, 70, 105, and 140 s adaptation-duration conditions respectively.

All experiments were run using custom software written in Python (PsychoPy [30], http://www.psychopy.org/).

**Statistical analysis.** First-level analyses quantified the spatial bias and gain of participants' responses in each condition. The predictor variable was defined as the auditory stimulus azimuth, and the outcome variable was defined as the participants' perceived azimuth. Multivariate outlier removal was applied to each condition and participant independently according to a robust Mahalanobis distance metric [31] and as described previously [15]; an alpha level of $p < .01$ was used as the rejection criterion, leading to 4.46% of all trials being rejected. Data were then entered into a series of linear regression analyses for each subject and for each of the 24 conditions independently. Spatial bias and gain were parameterised by the model intercept and slope coefficients respectively.

Next, these parameters were entered into second-level analyses testing the differences between conditions. We calculated the pairwise differences (across participants) in regression coefficients between adapting audio-visual disparities (-20˚ > +20˚) for each of the 12 main conditions (3 fixation-conditions × 4 adaptation-durations). The pairwise differences in spatial bias (intercept) parameters provide an estimate of the magnitude of the VAE, whilst the pairwise differences in spatial gain (slope) parameters measure gain change. In the following analyses, VAE magnitudes and gain changes were each analysed separately as they pertain to different measures. VAE magnitudes and gain changes were each entered into a two-way repeated-measures ANOVA with main effects of fixation-condition (eye+head-, eye-, and head-consistent) and adaptation-duration (35, 70, 105, and 140s). A Greenhouse-Geisser adjustment for sphericity was applied for all effects [32]. Post-hoc tests comprised pairwise t-tests between levels of the fixation-condition factor (collapsing over adaptation durations), and polynomial contrasts of the adaptation-duration factor. Effect sizes for ANOVAs are reported using both partial eta-squared ($\eta_p^2$) and generalised eta-squared ($\eta_G^2$) [33, 34]. Effect sizes for t-tests are reported using Hedges' $g_{av}$, in which the mean of the pairwise differences is divided by the mean of each pair's standard deviation [35, 36]. In addition, Bayes factors were calculated for all main effects and interactions in the ANOVA via the *BayesFactor* R-package (https://cran.r-project.org/package=BayesFactor), following the methods of [37].

All statistical tests were two-tailed and utilised an alpha criterion of 0.05 for determining statistical significance. Where applicable, a Holm-Bonferroni [38] correction for multiple comparisons was applied.

**Deviations from pre-registration.** We would like to note the following deviations from our pre-registered design plan:

1. The first-level analyses employ simple linear regression models within each participant individually, whereas the pre-registered plan proposed employing mixed-effects regression models with the participants entered as a random-effects factor. This approach would have entailed extracting per-participant mixed-effects parameter estimates for further statistical analysis. However, these estimates can be biased towards to the mean, which would be problematic for comparisons between the first and second experiments where sample sizes differ. Nevertheless, the mixed-effects regression models (S5 Fig) produced near identical parameter estimates to the updated simple regression model procedure (Fig 2).

2. The second-level repeated-measures ANOVA analysis was proposed to include all three levels of the fixation-condition factor (eye+head-, eye-, and head-consistent). However, this would mean the fixation-condition by adaptation-duration interaction would be contaminated by the influence of the eye+head-consistent condition, whilst our hypothesis more critically depends on just the eye- and head-consistent conditions. To address this, we repeated the ANOVA as planned but with the eye+head-consistent condition removed. These analyses are reported in S2 Methods, and yielded largely similar results to the original analyses.

## Results

Participants' responses were parameterised by entering the data into a series of linear regression models for each participant and condition separately. These models provided good fits to the data (S1–S4 Figs). Participants' spatial bias and gain parameters in each condition were represented by the regression intercept and slope coefficients respectively (Fig 2a). Performing these regression analyses using mixed-effects models, entering participants as a random-effects factor, yielded near identical parameter estimates (S5 Fig). Next, coefficients across

 

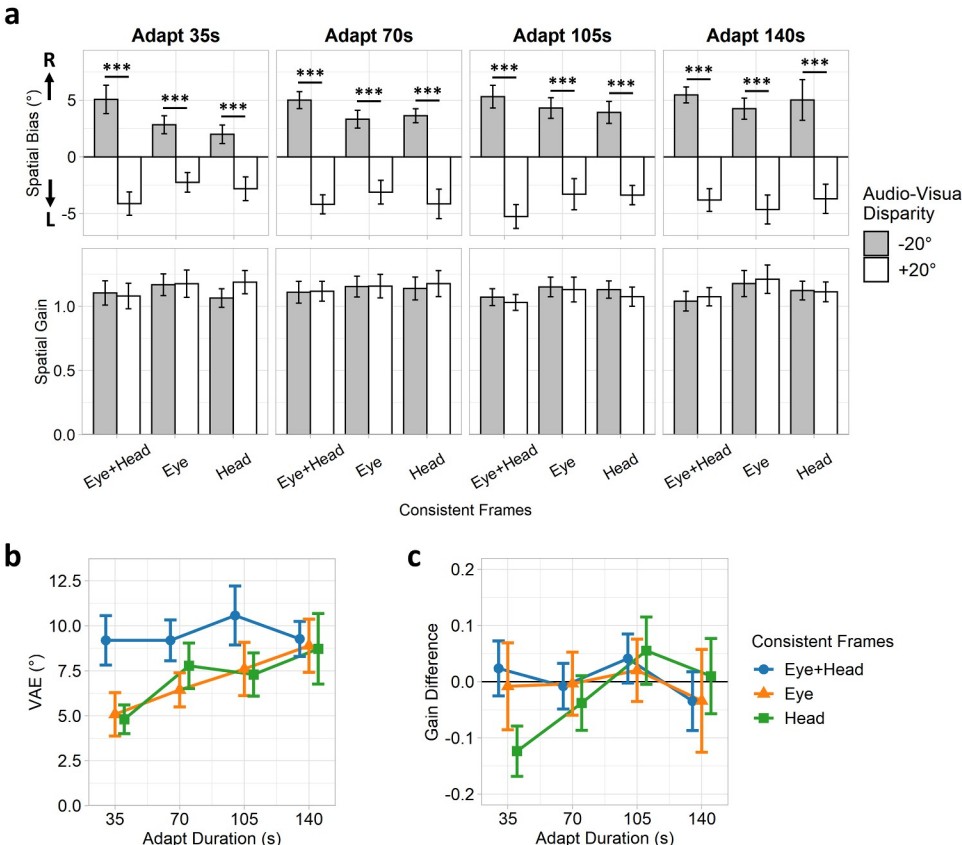

**Fig 2. Experiment 1: Group spatial bias and gain estimates.** (a) Spatial bias (intercept) and gain (slope) coefficients for each condition. Error bars indicate standard errors of the coefficients. (b) VAE magnitudes and (c) gain differences, quantified by contrasting spatial bias and gain coefficients between adaptation disparities (-20˚ > +20˚). Positive VAE magnitudes indicate spatial recalibration in the direction of the visual offset. Error bars indicate standard errors of the mean.

participants were contrasted between the adaptation disparities (-20˚ > +20˚) for each fixation condition and adaptation duration separately. Spatial bias differences thus indicate the magnitude of the VAE, with positive values indicating a spatial recalibration in the expected direction of the visual stimulus offset. Spatial gain differences meanwhile are not expected to differ substantially from zero or between conditions.

We first tested the VAE magnitude estimates (Fig 2b). A series of one-sample t-tests revealed that VAE magnitudes were significantly greater than zero in all conditions (all $p <$ .001, Holm-Bonferroni corrected), indicating spatial recalibrations in the direction of the visual adaptation offset. This confirmed that a VAE was elicited in all adaptation conditions. Next, VAE magnitudes were submitted to a two-way repeated measures ANOVA with factors of fixation-condition (eye+head-, eye-, head-consistent) and adaptation-duration (35, 70, 105, 140s). There was a significant main effect of fixation-condition (F(1.94, 36.78) = 7.05, $p$ = .003, $\eta_P^2 = .27$, $\eta_G^2 = .04$), supported by the equivalent Bayesian ANOVA which indicated substantial support for the alternative hypothesis ($BF_{10}$ = 15.65). A series of pairwise t-tests (collapsing over adaptation durations) indicated this main effect was due to higher VAE magnitudes in the eye+head-consistent than the eye-consistent (t(19) = 3.13, $p$ = .017, Hedges' $g_{av}$ = 0.62, $BF_{10}$ = 8.43) and head-consistent conditions (t(19) = 3.12, $p$ = .017, Hedges' $g_{av}$ = 0.55, $BF_{10}$ = 8.26), whilst the eye- and head-consistent conditions themselves did not differ significantly

($t(19) = 0.24$, $p = .814$, Hedges' $g_{av} = 0.05$, $BF_{10} = 0.24$). The main effect of adaptation-duration approached significance ($F(2.35, 44.68) = 2.88$, $p = .058$, $\eta_P^2 = .13$, $\eta_G^2 = .03$), although the Bayesian ANOVA did not indicate conclusive support either way ($BF_{10} = 0.84$). Post-hoc poly-nomial contrasts indicated this effect was mediated by a significant positive linear trend across adaptation durations ($t(57) = 2.85$, $p = .006$), whilst higher-order contrasts were not significant (quadratic: $t(57) = 0.72$, $p = .473$; cubic: $t(57) = 0.12$, $p = .902$). Critically, if the reliance of the VAE on eye- versus head-centred visual reference frames changes across time, then an interac-tion would be expected between fixation-condition and adaptation-duration. Contrary to our hypothesis, the fixation-condition by adaptation-duration interaction was not significant ($F(3.89, 73.97) = 0.88$, $p = .479$, $\eta_P^2 = .04$, $\eta_G^2 = .02$), and indeed the Bayesian ANOVA indicated substantial support for the null hypothesis ($BF_{10} = 0.06$). Repeating these analyses with the eye +head-consistent condition removed yielded largely similar results, with the exception that the main effect of fixation-condition was no longer significant (see S2 Methods). Similarly, repeat-ing the analyses for each of the four intra-block adapt/test cycles separately produced substan-tially similar results and did not indicate any significant effects of the cycle number (S6a Fig). Thus, although the VAE could be elicited using either eye- or head-centred visual frames, the relative contributions of each of these frames did not change with adaptation duration.

Finally, we considered the spatial gain differences. A series of one-sample t-tests indicated that the spatial gain differences did not differ significantly from zero ($p = .149$ for head-consis-tent, 35 s condition; all other $p > .999$; Holm-Bonferroni corrected). A repeated-measures ANOVA revealed no significant main effect of fixation-condition ($F(1.84, 34.88) = 0.43$, $p = .638$, $\eta_P^2 = .02$, $\eta_G^2 < .01$, $BF_{10} = 0.06$) or adaptation-duration ($F(1.90, 36.14) = 0.74$, $p = .480$, $\eta_P^2 = .04$, $\eta_G^2 = .01$, $BF_{10} = 0.08$), and no significant interaction ($F(3.97, 75.39) = 0.72$, $p = .579$, $\eta_P^2 = .04$, $\eta_G^2 = .02$, $BF_{10} = 0.05$). Repeating these analyses with the eye+head-consistent condition removed yielded substantially similar results (see S2 Methods). Thus, adaptation did not alter participants' spatial gain, and these estimates did not differ between conditions.

## Discussion

In our first experiment we tested the relative contributions of eye- and head-centred visual sig-nals to the VAE. We presented spatially disparate audio-visual pairs, but with the auditory stimulus located relative to either the head position (eye-consistent) or to the location of the fixated visual stimulus (head-consistent). In this way, a constant audio-visual disparity would be experienced for visual signals taken from one reference space (eye- or head-centred), while disparities would be variable for visual signals taken from the other reference space. Assuming that a robust VAE relies on adapting to a consistent audio-visual disparity, then visual signals from the reference space consistent with the auditory stimulus locations would be expected to be the primary driver of spatial recalibration effects.

We observed significant VAEs in all fixation conditions, with little difference between the eye- and head-consistent conditions. Consistent with [25], this suggests that both eye- and head-centred visual signals contribute to the spatial recalibration effects and in approximately equal magnitudes. We also observed significant VAEs across all durations tested. Whilst VAE magnitudes increased following longer adaptation periods, contrary to our hypothesis we did not find any evidence for an interaction between adaptation duration and spatial reference frames. We previously demonstrated that spatial recalibration effects can be underpinned by mechanisms tuned to both shorter and longer timescales [15]; our current results therefore suggest that such mechanisms rely on a common set of visual reference frames.

A key assumption underlying our methodology is that the VAE will be disrupted following exposure to spatially inconsistent (relative to consistent) audio-visual disparities. However,

recent evidence suggests that comparable VAE magnitudes may be elicited with either constant or variable audio-visual disparities [39]. This raises the possibility that our manipulation of the audio-visual spatial consistency may have had limited effect on the VAE. In this case, both eye- and head-centred visual signals may have contributed to the VAE in all conditions, thereby obfuscating inferences about the contributions of specific reference spaces. The disruption of the VAE in the eye- and head-consistent conditions (relative to the eye+head-consistent condition) may instead have reflected other task differences, such as the requirement to make eye movements. On the other hand, the magnitude of variability in audio-visual disparities is greater in our experiment than in [39], and indeed includes some disparities crossing the zero-point, and thus may be expected to have a more deleterious effect on the VAE. It is therefore necessary to provide a direct test for the effects of audio-visual spatial consistency within the context of our experimental paradigm. To this end, we conducted a second experiment reproducing the design of the eye+head-consistent condition from the first experiment, only now comprising variable spatial disparities. Equivalent audio-visual disparities are yielded by both eye- and head-centred visual frames, but the magnitude of those disparities varies over trials, reproducing the variability obtained using visual signals from the *inconsistent* reference spaces in the first experiment's eye- and head-consistent conditions. This effectively provides an eye+head-*inconsistent* condition. If spatial consistency has little effect on the VAE, then we would expect to find equivalent VAE magnitudes to the eye+head-consistent condition of the first experiment. However, if our original assumption holds, and disrupting the spatial consistency does affect the VAE, then we would instead expect VAE magnitudes to be reduced relative to the eye+head-consistent condition of the first experiment.

## Experiment 2: The effect of spatial consistency

### Methods

**Participants.**   Twelve participants took part in the study (6 male, 6 female, median age = 32, age range = 24–39). The study was approved by the ethics committee of the School of Psychology at the University of Nottingham (ethics approval number: 902) and conducted in accordance with the guidelines and regulations of this committee and the Declaration of Helsinki. Participants provided informed written consent before participating in the study.

The sample size was determined by an *a priori* power analysis. The critical comparison for detecting the VAE amounts to a paired-samples t-test between the average adapting audio-visual disparities (-20˚ versus +20˚). Note that for convenience we in practice conduct this as a one-sample t-test on the pairwise differences. We obtained a lower-bound estimate of the expected effect size from the equivalent comparisons in Experiment 1. The smallest effect size obtained in any comparison was Cohen's $d_z = 0.94$ (in the 35s eye-consistent condition). Entering this into G$^*$Power (v3.1; [27]) revealed that a minimum sample size of 11 would be required to obtain 80% power for a paired-sample t-test with an alpha criterion of 0.05. We therefore increased the sample size slightly to 12.

**Materials, design, and procedure.**   The experimental materials, design, and procedure followed those of the eye+head-consistent condition in the first experiment, with the only modification that audio-visual disparities were now variable over trials. Participants oriented their head directly forward (assisted by a chin-rest), and maintained central fixation throughout. Auditory stimuli were always presented relative to the visual stimulus location. Thus, both eye- and head-centred visual reference frames yielded the same audio-visual disparities. A within-subjects design was employed with two main factors: adaptation-duration (35, 70, 105, 140 s) and mean audio-visual disparity (±20˚).

During adaptation phases, participants were presented with synchronous audio-visual stimulus pairs. Visual stimulus locations ranged between -30˚ (left) and +30˚ (right) eccentricity in 10˚ increments (7 eccentricities total). Auditory stimulus locations were sampled from a uniform range with a mean of either -20˚ (leftward shift) or +20˚ (rightward shift) offset from the visual stimulus, but varied over trials by up to ±30˚ either side of the mean in 10˚ steps (7 auditory offsets total). Thus, auditory stimuli could be offset between -50˚ and +10˚ (−20˚ mean offset) or between -10˚ and +50˚ (+20˚ mean offset) of the visual stimulus location. This reproduces the variability obtained from the *inconsistent* visual reference space in the eye- and head-consistent conditions in the first experiment (Fig 1b). Each adaptation phase comprised 1, 2, 3, or 4 passes (35, 70, 105, or 140 s) over all visual locations (in a randomised order) according to the adaptation-duration condition of the block. Within each pass, each of the 7 possible auditory offsets were used once in a random order. In this way, audio-visual disparities were distributed randomly and uniformly throughout each adaptation phase. A schematic illustration of an example trial is provided in Fig 3a, and the distributions of audio-visual disparities are shown in Fig 3b. All other experimental details, including those for the test phases, are the same as for the first experiment.

Conditions were presented within a blocked-design, with each of the 8 conditions (4 adaptation-durations × 2 spatial-disparities) allocated to one block of the experiment. The block order was fully randomised for each participant. Each block comprised 4 cycles of alternating adaptation and test phases. Average block durations were 3m43s, 6m3s, 8m25s, and 10m44s for the 35, 70, 105, and 140 s adaptation-duration conditions respectively.

**Statistical analysis.** The first-level analysis proceeded as per the first experiment. Multivariate outlier removal [31] was performed as described above (leading to the rejection of 4.22% of all trials), and data were entered into a series of linear regression analyses for each subject and condition separately. Spatial bias and gain were parameterised by the model intercept and slope coefficients respectively.

Outputs were then entered into second-level analyses testing the differences between conditions. Spatial bias (intercept) and gain (slope) parameters were analysed separately. We calculated pairwise differences in the parameters between audio-visual disparities (-20˚ > +20˚). A series of one-sample t-tests contrasted these differences against zero, subject to a Holm-Bonferroni [38] correction for multiple comparisons over the adaptation durations. Next, the difference values were entered into a series of ANOVAs. We first employed a one-way repeated-measures ANOVA to test the main effect of adaptation duration (35, 70, 105, and 140 s). To examine the difference between constant and variable audio-visual disparities, we then further compared the second experiment to the equivalent eye+head-consistent condition from the first experiment using a two-way mixed-design ANOVA with a between-subjects factor of disparity-type (constant or variable) and a repeated-measures factor of adaptation-duration. All ANOVAs employed a Greenhouse-Geisser correction for sphericity [32], and both partial and generalised eta-squared effect sizes are reported [33, 34]. In addition, Bayes factors are reported using the *BayesFactor* R-package (https://cran.r-project.org/package=BayesFactor). All statistical tests were two-tailed and utilised an alpha criterion of 0.05 for determining statistical significance.

## Results

Linear regression models fit to each participant's responses for each condition again provided a good fit to the data (S7 Fig). Spatial bias and gain parameters were quantified by the intercept and slope coefficients of the regression models (Fig 3c and 3d). Near identical parameter estimates were also obtained using mixed-effects models entering the participants as a random-

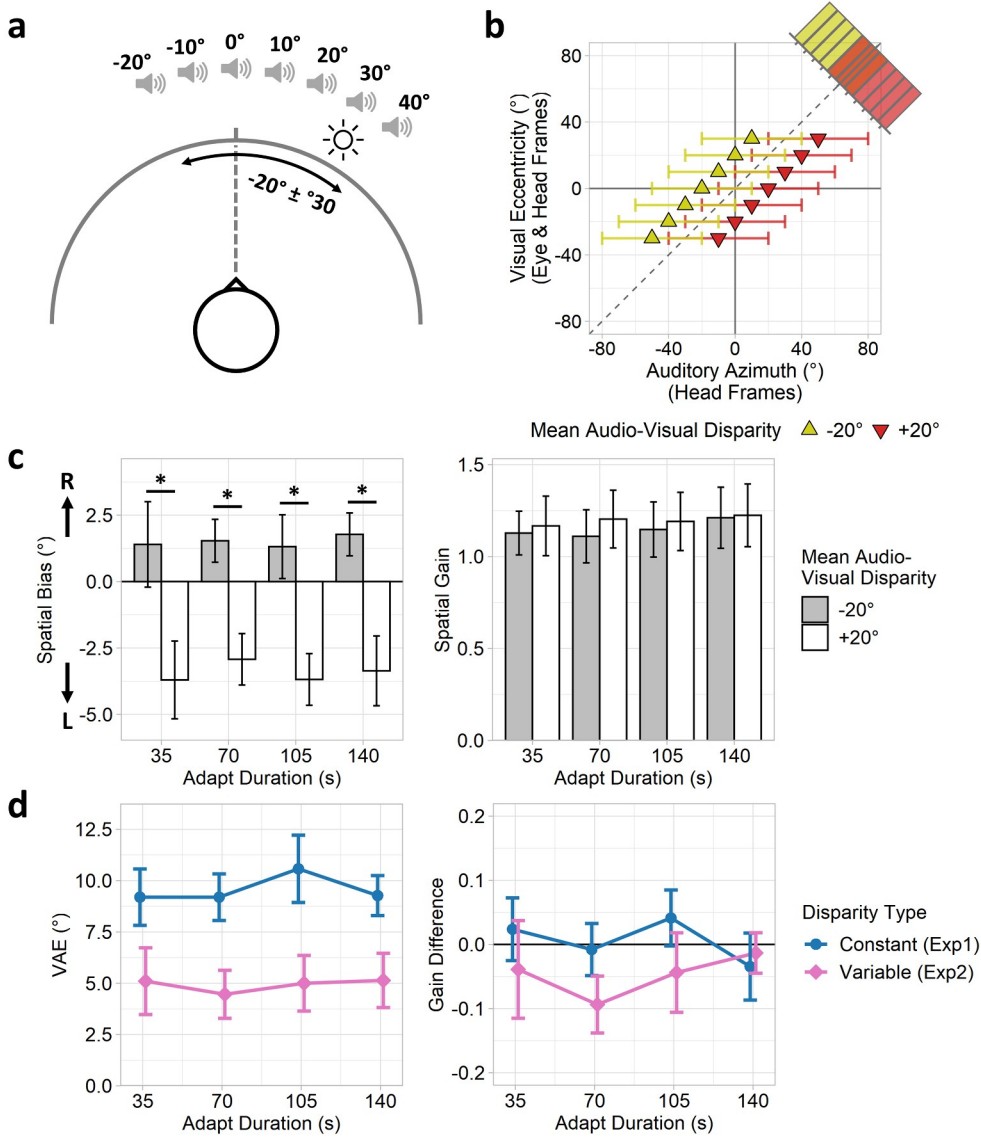

**Fig 3. Experiment 2: Design, and group spatial bias and gain estimates.** (a) Schematic illustration of an example trial pairing a visual stimulus at +30˚ eccentricity with an auditory stimulus positioned ±30˚ of an average -20˚ leftward offset. (b) Auditory azimuth plotted against visual eccentricity. Eye- and head-centred visual signals yield equivalent audio-visual disparities but which vary over trials. Markers and error bars indicate means and ranges of possible auditory locations for each visual location. Corner histograms illustrate distributions of audio-visual disparities. (c) Spatial bias (intercept) and gain (slope) coefficients for each condition. Error bars indicate standard errors of the coefficients. (d) VAE magnitudes and gain differences, quantified by contrasting spatial bias and gain coefficients between adaptation disparities (-20˚ > +20˚). Values are shown for both constant (blue; Experiment 1: eye+head-consistent condition) and variable disparities (pink; Experiment 2). Positive VAE magnitudes indicate spatial recalibration in the direction of the mean visual offset. Error bars indicate standard errors of the mean.

effects factor (S8 Fig). Next, coefficients were contrasted between the adaptation disparities (-20˚ > +20˚) for each adaptation duration in turn, and these values were submitted to further analyses. Spatial bias differences indicate the magnitude of the VAE.

We first tested whether VAEs were still present using a series of one-sample t-tests to contrast VAE magnitudes against zero. These revealed significant effects for all adaptation-durations (all $p = .010$, Holm-Bonferroni corrected), indicating VAEs were indeed elicited even

with the variable audio-visual disparities. Next, we performed a one-way repeated-measures ANOVA to test the effect of adaptation duration. This found no significant effect of duration (F(1.85, 20.35) = 0.07, $p$ = 925, $\eta_P^2 < .01$, $\eta_G^2 < .01$), and the equivalent Bayesian ANOVA indicated substantial support for the null hypothesis ($BF_{10}$ = 0.12). Thus, VAEs did not increase across the adaptation durations.

We also tested the effect of constant versus variable audio-visual disparities using a mixed-design ANOVA to compare VAEs from the second experiment (variable disparities) against those from the corresponding eye+head-consistent condition in the first experiment (constant disparities). This revealed a significant between-subjects effect of disparity-type (F(1,30) = 9.89, $p$ = .004, $\eta_P^2 = .25$, $\eta_G^2 = .15$, $BF_{10}$ = 11.01) due to larger VAE magnitudes with constant than variable disparities. The within-subjects effect of adaptation-duration was again non-significant (F(2.44, 73.17) = 0.29, $p$ = .788, $\eta_P^2 < .01$, $\eta_G^2 < .01$, $BF_{10}$ = 0.07), as was the disparity-type by adaptation-duration interaction (F(2.44, 73.17) = 0.26, $p$ = .812, $\eta_P^2 < .01$, $\eta_G^2 < .01$, $BF_{10}$ = 0.13). Repeating these analyses for each of the four intra-block adapt/test cycles separately produced substantially similar results and did not indicate any significant effects of the cycle number (S6b Fig). Thus, although VAEs were still elicited following adaptation to variable audio-visual disparities, they were nevertheless significantly reduced in magnitude compared to the equivalent constant disparities.

Finally, we examined the spatial gain differences. One-sample t-tests did not find a significant difference from zero for any adaptation duration ($p$ = .178 for 70 s condition; all other $p$ >.999; Holm-Bonferroni corrected). A one-way repeated-measures ANOVA showed no significant effect of adaptation-duration (F(2.33, 25.62) = 0.40, $\eta_P^2 = .03$, $\eta_G^2 = .02$, $BF_{10}$ = 0.17). Similarly, a mixed-design ANOVA comparing spatial gain differences with the first experiment revealed no significant effect of disparity-type (F(1,30) = 2.68, $p$ = .111, $\eta_P^2 = .08$, $\eta_G^2 = .02$, $BF_{10}$ = 0.47), adaptation-duration (F(2.65, 79.43) = 0.38, $p$ = .741, $\eta_P^2 = .01$, $\eta_G^2 < .01$, $BF_{10}$ = 0.08), or their interaction (F(2.65, 79.43) = 0.47, $p$ = .680, $\eta_P^2 = .02$, $\eta_G^2 = .01$, $BF_{10}$ = 0.16). Thus, spatial gain was unaffected by adaptation.

## Discussion

Our second experiment aimed to directly test the effect of adapting to constant versus variable audio-visual disparities on the ventriloquism aftereffect. This is critical to the interpretation of our first experiment, which was predicated on the assumption that spatial recalibration effects should be primarily driven by visual signals from reference spaces spatially consistent with audio signals. Contrary to this assumption, recent evidence actually suggests that equivalent recalibration effects may be obtained with either constant or variable spatial disparities [39].

We reproduced the eye+head-consistent condition from the first experiment, such that both eye- and head-centred visual signals yielded equivalent audio-visual disparities, only now those disparities varied over trials. Crucially, the range of disparities was selected to reproduce the variability obtained from the inconsistent visual reference space in the eye- and head-consistent conditions. Consistent with [39], significant VAEs were still observed in spite of the variable audio-visual disparities. Importantly, however, VAEs were also significantly reduced in magnitude compared to those produced by the equivalent constant disparities in the first experiment. This indicates that the recalibration effects observed in the first experiment would have been primarily driven by visual signals from the *consistent* reference space, although signals from the inconsistent space may still have made a limited contribution. It is therefore very unlikely that the differences observed between conditions within the first experiment could have been primarily driven by factors other than the spatial consistency of the audio-visual

disparities (such as task differences). Instead, the results are supportive of the VAE operating in both eye- and head-centred reference spaces.

## General discussion

In this study we tested the relative contributions of eye- and head-centred visual reference frames to the ventriloquism aftereffect (VAE) across different timescales of adaptation ranging from a few seconds to a few minutes. In our first experiment, we manipulated the position of the auditory stimulus relative to either the position of the head or of the fixated visual stimulus, allowing us to selectively maintain or disrupt the spatial consistency between audio-visual pairs given by either eye- or head-centred visual reference spaces. A control condition in which participants maintained central fixation throughout provided a further measure of combined eye- and head-centred visual frames. Consistent with previous research [25], we found the VAE could be elicited from both eye- and head-centred visual frames, such that the VAE remained evident when spatial consistency within one reference frame was disrupted yet maintained within the other. However, contrary to our hypothesis that the fixation condition and adaptation duration would interact, we did not find any indication of a shift between reference frames across different timescales of adaptation. A follow-up experiment then confirmed that the manipulation of audio-visual spatial consistency did modulate the magnitude of the VAE.

In our first experiment, a reliable VAE was elicited in all fixation conditions and across all the tested adaptation durations. The VAE was strongest when both eye- and head-centred visual frames were consistent with the auditory locations, and was disrupted when only eye- or head-centred visual frames were consistent (though comparable between those cases). The VAE magnitude increased across adaptation durations: this effect was most evident in the eye- and head-consistent conditions, whilst the VAE appeared mostly saturated from the shortest adaptation duration (35s) onwards for the eye+head-consistent condition. Indeed, previous studies have reported near saturation of the VAE within a few tens of seconds of adaptation [10, 15]. VAE magnitudes appeared comparable between all fixation conditions following the longest adaptation duration (140s). This indicates that disrupting the consistency of audio-visual disparities from either eye- or head-centred visual frames leads to a delay in the saturation point of the VAE, such that longer durations of adaptation are required for the VAE to reach the same saturated level achieved after brief exposures when both reference frames yield consistent disparities. The fact that VAEs can be elicited via head-centred visual reference frames also implies that paradigms aiming to elicit audio-visual recalibration need not necessarily control participants' eye movements, and indeed many previous studies have not employed such controls.

Overall, these results confirm that the ventriloquism aftereffect is underpinned by both eye- and head-centred visual frames, and replicate the findings of Kopčo and colleagues [25]. However, a more recent replication study by the same group found strong evidence for head-centred representations, but little evidence for eye-centred influences [26]. The methodology employed by Kopčo and colleagues measures spatial tuning curves around VAEs elicited within a restricted range of azimuths: their former and latter experiments differ in that adaptation was elicited in central versus peripheral ranges respectively, suggesting recalibration effects may be less eye-centred in the periphery. However, our experiment found relatively uniform recalibration effects across space using both eye- and head-centred visual frames: spatial gain did not differ between conditions (Fig 2a and 2c), and model residuals appeared relatively evenly distributed across azimuths in all conditions (S1b, S2b, S3b, S4b and S7b Figs). Thus, our results suggest VAEs can be elicited from both eye- and head-centred visual

reference frames in both central and peripheral locations. This discrepancy may reflect that our adaptation included both central and peripheral locations, whereas [26] restricted adaptation to just peripheral azimuths. Furthermore, our paradigm estimates a global spatial bias across a range of azimuths, whereas Kopčo and colleagues' paradigm requires measuring biases at each individual azimuth and is thus inherently dependent on the quality of the measured spatial tuning curves.

Whilst our method disrupts the consistency of audio-visual disparities yielded by a given visual reference frame, it does not eliminate spatial biases in this frame. As illustrated by the corner histograms in Fig 1b, the distribution of audio-visual disparities using the inconsistent visual space (visual eye frames for the head-consistent condition, visual head frames for the eye-consistent condition) retains an average leftward or rightward bias depending on the adapting audio-visual disparity. Thus, a VAE could potentially still be elicited from the inconsistent visual reference space. Indeed, recent evidence by Bruns and colleagues [39] suggests that VAEs can be reliably elicited even from variable audio-visual disparities. To test this possibility, we conducted a follow-up experiment which reproduced the eye+head-consistent condition from the first experiment, such that both eye- and head-centred visual signals yielded equivalent audio-visual disparities, but which varied over trials so as to reproduce the variability obtained from visual signals under the *inconsistent* reference space of the eye- and head-consistent conditions. This effectively provided an *eye+head-inconsistent* condition. Significant VAEs were obtained in spite of the variable disparities, however they were reduced in magnitude compared to those obtained with constant disparities in the equivalent condition from the first experiment. This demonstrates that the recalibration effects observed in the first experiment were primarily driven by visual signals from the *consistent* reference space, though signals from the inconsistent space may have made lesser contributions. This therefore supports the conclusions of the first experiment: both eye- and head-centred visual signals contribute to the VAE.

Unlike the eye- and head-consistent conditions in the first experiment, the VAEs in the second experiment never recovered to the level of the eye+head-consistent condition, even after the longest adaptation duration. Disrupting the consistency of audio-visual disparities in both eye- and head-centred frames may yield a more global reduction in the VAE magnitude across all adaptation durations, though it may also have simply delayed the saturation point beyond our longest adaptation period and the VAE would have eventually recovered with further exposure. The results of the second experiment also partially diverge from those of Bruns and colleagues: whilst we also find evidence for VAEs elicited by variable disparities, they found equivalent magnitudes of VAEs between constant and variable disparities, whereas we found stronger effects with constant disparities. This discrepancy could reflect methodological differences; for instance, Bruns and colleagues employed longer adaptation periods than we did (5 minutes, relative to 2 minutes 20 seconds in our longest condition). Furthermore, our experiment presented a wider range of spatial disparities (up to ±30˚ around the mean disparity, including disparities crossing the zero point) compared to Bruns and colleagues who only presented disparities up to ±8.1˚ around the mean and which didn't cross the zero point.

Visual signals are acquired in eye-centred co-ordinates, whilst auditory signals are acquired in head-centred frames. The comparable VAE magnitudes we observed in the eye- and head-consistent conditions demonstrate sensory recalibration can be accomplished using either eye- or head-centred visual signals. It is easy to see how spatiotopic transformations of visual signals [18] could be combined with head-centred auditory signals, but it is less obvious how the original eye-centred visual signals should be integrated. One possibility is eye-centred visual signals are directly combined with head-centred auditory signals, such that the visual fovea is mapped to 0˚ auditory azimuth regardless of eye position. Such an approach seems counterintuitive, as

multi-modal objects in naturalistic settings are typically spatially consistent in world-centred rather than eye-centred coordinates. Speculatively, sensory recalibration in this manner could help correct for audio-visual disparities arising from eye movements. Indeed, some auditory spatial receptive fields at both sub-cortical [40–42] and cortical [43] sites are modulated by eye-movements, and fixations may induce gradual perceptual shifts of auditory space [44], indicating some precedence for the integration of eye-centred visual signals. Alternatively, following a direct combination of visual eye-centred and auditory head-centred signals, any corresponding recalibration effects may simply reflect an inevitable consequence of the system architecture. Similar findings have been reported in the somatosensory literature, in which tactile inputs may be combined with signals from other modalities without remapping the co-ordinate frames under certain conditions [45–48].

Our paradigm assumes that while visual signals can be represented in eye- or head-centred frames, auditory signals are only encoded in head-centred frames. An alternative possibility is that auditory signals could be transformed into eye-centred co-ordinates, which would alter the interpretation of our findings. However, we feel that such an account is unlikely for a number of reasons. Firstly, while some auditory spatial receptive fields are modulated by eye movements [40–43] as mentioned above, this does not appear to yield completely eye-centred representations. Meanwhile, there is clear evidence for spatiotopic transformations of visual signals to head-centred co-ordinates [18]. Secondly, perceptual shifts in auditory space caused by fixation changes have been shown to develop gradually over several seconds or minutes [44], yet in our paradigm the fixation position updated to a new random location at a much faster timescale. Finally, in our eye-consistent condition, an eye-centred transformation of the auditory signals would have rendered them spatially inconsistent with both eye- and head-centred visual signals. Whilst the visual stimulus remained at the same foveal eye-centred position, the auditory stimulus location (fixed relative to the head) would vary in eye-centred co-ordinates as the fixation position changed. Conversely, head-centred visual signals would change in opposite directions to an eye-centred representation of the auditory location; for example, fixating a visual stimulus to the left of the head would place the auditory location in the right visual field. It would therefore be difficult to explain the robust VAE elicited in this condition, or how the VAE could be largely equivalent in magnitude between the eye- and head-consistent conditions. Thus, our results are most consistent with mixed contributions of eye- and head-centred visual signals, but with auditory signals remaining head-centred.

The primary aim of this study was to test for differences in the relative contributions of eye- and head-centred visual frames to the VAE across different timescales. Contrary to our hypothesis, we did not find any evidence for an interaction in the VAE between the fixation condition and adaptation duration, and indeed Bayesian tests indicated substantial support for the null hypothesis. This is also in contradiction to the supplementary analyses of Kopčo and colleagues [25], which suggested a transition from head- to eye-centred visual representations with an increasing duration of adaptation. The longest adaptation duration we tested (140s) falls substantially shorter than that of Kopčo and colleagues, who used extended adaptation periods comprising over 700 trials (approximately 30 minutes); thus, a clearer eye-centred advantage may have emerged if we had included longer periods of adaptation. Nevertheless, even following 140 seconds of adaptation, the VAE in both the eye- and head-consistent conditions appeared close to the magnitude of the eye+head-consistent condition, suggesting that the VAE under each reference frame was approaching saturation and would be unlikely to change substantially with further adaptation. Equally well, a clearer head-centred advantage might have been apparent following shorter durations of adaptation than the shortest period included here (35 s). Indeed, rapid spatial recalibration effects have been reported following very short periods of adaptation of just a few seconds or even a single trial [12–14]. Testing

shorter durations would require a radical redesign of our current paradigm, which relies on presenting repeated adapting stimuli across a range of azimuths before each test phase (35 seconds is the time required to complete one full sweep of all azimuths). Nevertheless, 35 seconds of adaptation remains considerably shorter than the initial period tested by Kopčo and colleagues, which comprised the first quarter of each block, so the lack of a head-centred advantage in our study here remains a discrepant result. It is important to note that Kopčo and colleagues did not primarily aim to investigate the effect of adaptation duration and hence their study design was not optimised to test this hypothesis. By comparison, this was a direct aim of our study and our paradigm was optimised accordingly by explicitly employing different periods of adaptation.

In a previous study, we demonstrated that the VAE is supported by multiple distinct recalibration mechanisms operating over different timescales [15], consistent with other studies supporting a multiple-mechanisms account of the VAE [11, 13]. A key question is whether these different mechanisms rely on common or distinct visuo-spatial reference frames. In the current study we failed to find any evidence of a change in the visual reference frames underpinning the VAE with varying durations of adaptation, and thus our results suggest a reliance on common visual reference frames across mechanisms. Neuroimaging studies have implicated both primary auditory cortices [21–23] and multisensory parietal regions [24] in both immediate and sustained audio-visual spatial recalibration. Eye- and head-centred visual influences on the VAE could conceivably be supported by the concurrent retinotopic and spatiotopic visual representations in the parietal dorsal stream [18], and indeed some such regions also display overlapping visual and auditory receptive fields [20]. Alternatively, multiple visual reference frames could contribute to the VAE by an interplay between multiple sensory regions variously coding eye- or head-centred visual frames. The absence of an interaction between reference frames and adaptation duration in the VAE could indicate that both shorter- and longer-term recalibration mechanisms depend on a common neural substrate, but could equally well imply distinct neural substrates which simply do not employ differentiable frames of reference. Other studies have reported that the extent to which the VAE generalises over auditory frequencies depends on the duration of adaptation [13, 49], thus frequency-dependence may offer an alternative target for differentiating between recalibration mechanisms.

## Conclusion

This study examined the contribution of different visuo-spatial reference frames to the ventriloquism aftereffect and how these interact with the temporal scale of adaptation. In line with previous research, we found support for both eye- and head-centred visual contributions to audio-visual spatial recalibration. However, we found no evidence for an interaction between adaptation duration and reference frames. These results suggest that different recalibration mechanisms operating over distinct timescales rely on common spatial reference frames.

## Supporting information

**S1 Methods. Experiment 1: Adjusted power calculations.**
(DOCX)

**S2 Methods. Experiment 1: Second-level analyses without eye+head-consistent condition.**
(DOCX)

**S1 Fig. Experiment 1: Linear regression fits for 35s adaptation.** (a) Participants' perceived stimulus azimuth plotted against actual stimulus azimuth, following 35 seconds of adaptation to audio-visual pairs spatially offset by -20˚ (leftward audio offset; top row) or +20˚ (rightward

audio offset; bottom row), and for each fixation condition (eye+head-, eye-, and head-consistent; across columns). Data were entered into a series of linear regression analyses for each participant and condition separately. (b) Corresponding model residuals. Data points and model fits are colour-coded by participant.
(TIF)

**S2 Fig. Experiment 1: Linear regression fits for 70s adaptation.** As per S1 Fig, but following 70 seconds of adaptation.
(TIF)

**S3 Fig. Experiment 1: Linear regression fits for 105s adaptation.** As per S1 Fig, but following 105 seconds of adaptation.
(TIF)

**S4 Fig. Experiment 1: Linear regression fits for 140s adaptation.** As per S1 Fig, but following 140 seconds of adaptation.
(TIF)

**S5 Fig. Experiment 1: Mixed-effects regression analyses.** Parameter estimates obtained from mixed-effects regression models allowing random intercepts and slopes over participants. Results appear near-identical to those of the standard regression analyses (cf. Fig 2). (a) Spatial bias (intercept) and gain (slope) coefficients for each condition. Error bars indicate standard errors of the coefficients. (b) VAE magnitudes and (c) gain differences, quantified by contrasting spatial bias and gain coefficients between adaptation disparities (-20˚ > +20˚). Positive VAE magnitudes indicate spatial recalibration in the direction of the visual offset. Error bars indicate standard errors of the mean.
(TIF)

**S6 Fig. VAE estimates for each intra-block adapt/test cycle.** (a) Experiment 1 estimates. A three-way repeated-measures ANOVA revealed a significant main effect of fixation-condition ($F(1.62, 30.69) = 5.04$, $p = .018$, $BF_{10} = 1519.74$), but no significant main effects of adaptation-duration ($F(2.26, 42.93) = 2.11$, $p = .128$, $BF_{10} = 2.11$) or cycle-number ($F(2.73, 51.81) = 2.10$, $p = .117$, $BF_{10} = 0.01$). No interactions were significant (all $p > .05$, all $BF_{10} < 0.33$). (b) Experiment 2 estimates. A three-way mixed-design ANOVA revealed a significant main effect of disparity-type ($F(1, 30) = 9.82$, $p = .004$, $BF_{10} = 9.53$), but no significant main effects of adaptation-duration ($F(2.48, 74.29) = 0.16$, $p = .896$, $BF_{10} = 0.01$) or cycle-number ($F(2.25, 67.36) = 1.76$, $p = .175$, $BF_{10} = 0.05$). No interactions were significant (all $p > .05$, all $BF_{10} < 0.33$).
(TIF)

**S7 Fig. Experiment 2: Linear regression fits.** (a) Participants' perceived stimulus azimuth plotted against actual stimulus azimuth following adaptation to audio-visual pairs spatially offset by an average of -20˚ (leftward audio offset; top row) or +20˚ (rightward audio offset; bottom row). Adaptation durations are represented across columns. Data were entered into a series of linear regression analyses for each participant and condition separately. (b) Corresponding model residuals. Data points and model fits are colour-coded by participant.
(TIF)

**S8 Fig. Experiment 2: Mixed-effects regression analyses.** Parameter estimates obtained from mixed-effects regression models allowing random intercepts and slopes over participants. Results appear near-identical to those of the standard regression analyses (cf. Fig 3). (a) Spatial bias (intercept) and gain (slope) coefficients for each condition. Error bars indicate standard

errors of the coefficients. (b) VAE magnitudes and gain differences, quantified by contrasting spatial bias and gain coefficients between adaptation disparities (-20˚ > +20˚). Values are shown for both constant (blue; Experiment 1: eye+head-consistent condition) and variable disparities (pink; Experiment 2). Positive VAE magnitudes indicate spatial recalibration in the direction of the mean visual offset. Error bars indicate standard errors of the mean. (TIF)

## Author Contributions

**Conceptualization:** David Mark Watson, Michael A. Akeroyd, Neil W. Roach, Ben S. Webb.

**Data curation:** David Mark Watson.

**Formal analysis:** David Mark Watson.

**Funding acquisition:** Michael A. Akeroyd, Neil W. Roach, Ben S. Webb.

**Investigation:** David Mark Watson.

**Supervision:** Michael A. Akeroyd, Neil W. Roach, Ben S. Webb.

**Visualization:** David Mark Watson.

**Writing – original draft:** David Mark Watson, Michael A. Akeroyd, Neil W. Roach, Ben S. Webb.

**Writing – review & editing:** David Mark Watson, Michael A. Akeroyd, Neil W. Roach, Ben S. Webb.

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
