## [Decision Letter · Decision Letter 0]

19 Mar 2021

PONE-D-21-03641

Multiple spatial reference frames underpin perceptual recalibration to audio-visual discrepancies

PLOS ONE

Dear Dr. Watson,

Thank you for submitting your manuscript to PLOS ONE. After careful consideration, we feel that it has merit but does not fully meet PLOS ONE’s publication criteria as it currently stands. Therefore, we invite you to submit a revised version of the manuscript that addresses the points raised during the review process.

Please note that Reviewer 1 has provided a file with additional comments and a figure. 

We look forward to receiving your revised manuscript.

Kind regards,

Nicholas Seow Chiang Price, Ph.D.

Academic Editor

PLOS ONE

Journal Requirements:

Reviewers' comments:

Reviewer's Responses to Questions

**Comments to the Author**

1. Is the manuscript technically sound, and do the data support the conclusions?

Reviewer #1: Partly

Reviewer #2: Partly

2. Has the statistical analysis been performed appropriately and rigorously? 

Reviewer #1: Yes

Reviewer #2: Yes

3. Have the authors made all data underlying the findings in their manuscript fully available?

Reviewer #1: Yes

Reviewer #2: Yes

4. Is the manuscript presented in an intelligible fashion and written in standard English?

Reviewer #1: Yes

Reviewer #2: Yes

5. Review Comments to the Author

Reviewer #1: This study presents two experiments attempting to disentangle the contributions of eye-centric and head-centric reference frames to the VAE, and to determine whether the relative contributions of these reference frames changes with continued exposure to an AV offset. Experiment 1 contains the key manipulations attempting to break the consistency of the AV offset in each frame independently; Experiment 2 is a control study designed to test what influence such inconsistency in AV offset has on the strength of the VAE. Overall, I found the paper clear and well-written, the data a valuable contribution, and the statistics thorough and above-board. I have some suggestions regarding terminology and the interpretation of some key results, but I think the addition of the second experiment strengthens the study overall. [As an aside, I was not a reviewer for the initial submission of this manuscript.]

Major issues

1) “Coherent AV disparity” is not a very clear term.

It took until I was well into the paper to understand what you meant by it. All your stimuli are in a sense “spatially incoherent” – the perceptual system is recalibrating to something perceived to have gone wrong, after all. I should also mention that in my corner of the multisensory perception field, coherence is a borderline reserved term referring to temporal coherence (e.g. the co-modulation of contrast and amplitude you used to encourage AV binding). I would suggest that you replace this term with “consistent AV disparity” (or similar) whenever possible to promote general readability.

2a) Is the VAE really “reduced” or “weakened” in Exp. 1?

In the EC and HC conditions, in which the consistent AV offset was disrupted in one of the two reference frames, the VAE was reduced when the adaptation duration was relatively short. However, in both of these conditions, the VAE eventually recovered to the same magnitude as the E+H condition by the longest adaptation duration. You do mention this point (e.g. L541), but to me, the terms “reduced” or “weakened” used elsewhere don’t accurately capture the data. In the EC and HC conditions, it’s not really that the magnitude of the VAE was reduced, but that it took more exposure to reach that maximum magnitude. This is in contrast to the results of Exp. 2, which showed a truly reduced VAE across all adaptation durations tested. Being more explicit about this difference might actually work to the benefit of your main claims (see next point).

2b) Could Exp. 2 be framed as breaking the consistency of the AV offset in both reference frames?

You claim that Exp. 2 is set up to parallel the E+H condition of Exp. 1 (e.g. L379). Here (and also at L502), you say that “both eye-and head-centered reference frames [were] matched.” It’s not entirely clear what is meant by “matched” here, but I think it means that the variability in AV offset was the same whether the visual stimuli are considered in EC or HC coordinates. But put another way, this means that there was no consistent AV disparity in either reference frame (i.e., as I understand it, this was an Eye+Head Incoherent condition). Unlike the EC and HC conditions of Exp. 1, now the VAE never recovered at any adaptation duration tested. If this description is valid, then what you have is a delayed VAE if consistency is disrupted in either the HC or EC reference frame (as if the system can recover from ambiguity in one frame), and a permanently reduced VAE if you disrupt the consistent AV offset in both frames (Exp. 2). In any case, a more explicit explanation of this result is important – why did the VAE eventually recover in the EC and HC conditions, but not in Exp. 2, despite your attempt to match the extent of variability in AV offset?

On the topic, I have a somewhat nitpicky issue with your framing of Exp. 2. As you know, the inconsistent frames in your EC and HC conditions include zero-disparity AV exposures, wrong-direction exposures, as well as exposures to AV offsets so large as to potentially fall outside the spatial window of visual influence over auditory perception. It would have been shocking if these factors did not reduce the VAE, but you needed to show conclusively that they do. All fine. But in your setup to Exp. 2 you write, for instance, “the precise magnitude of VAEs expected following adaptation to variable disparities remains unclear.” The implication is that you’ll be doing some careful manipulation to determine what types of variability affect the VAE, when in reality you’re smashing it with a hammer to make sure it breaks (as needed for compatibility with Exp. 1). I would appreciate this experiment being set up something more like:

• [Describe previous work investigating the effect of spatial variability (Bruns et al.)]

• However, in the EC and HC conditions of our study, variability in the inconsistent frames was much larger than what has been previously investigated, and included zero-crossings.

• We needed to measure whether this large extent of variability negatively impacts the VAE, as we assumed in our design.

3) Potential recalibration of auditory spatial perception in conditions with changing eye position could be more thoroughly addressed.

In a couple instances, you mention that auditory spatial receptive fields are known to shift with changes in eye position. Importantly, this also manifests in human behavior, with auditory spatial perception shifting by up to about 40% in the direction of eye gaze. The time constant of this process varies widely across participants, but averages on the order of a couple minutes (Razavi et al. J. Neuro, 2007).

Specifically, I’m wondering about this in the context of whether the visual stimulus locations were randomized within an adaptation cycle, or if they proceeded in a set order, e.g. left to right. [Unless I missed it, please clearly state which of these was the case in your methods.] If the visual stimulus was in the same hemifield for multiple consecutive stimuli, then although the drive to shift auditory space would change depending on stimulus eccentricity, it would be in the same direction and thus possibly have a compounding effect across stimulus presentations. This would systematically alter the perceived AV disparity, shrinking or magnifying it depending on hemifield and condition.

In reality, I think even if multiple consecutive adaptations occurred in the same hemifield, there wouldn’t be enough time for these eye position effects to drastically alter your results. Nonetheless, I would appreciate it if the possibility were addressed in the discussion.

4) I don’t think 10s between adaptation cycles should reset the VAE at all close to baseline.

In fact, during a relatively short period of auditory localization following exposure to an AV offset, the VAE doesn’t necessarily decline toward baseline at all:

[see attached version of review for image – it’s Fig 3 from the cited paper]

Blue = pre-exposure localization, grey = localization during AV exposure, red = post-exposure localization. Red period is roughly a couple minutes of auditory localization. From Bosen et al., PLoS one, 2018. [Full disclosure: I am a coauthor on this study, but you already cited it anyway.]

Because of this, I would guess that you actually have compounding adaptation across the 4 cycles within each block. One way to check for this (and a possible supplemental figure) would be to separately analyze the data from each cycle, to see if your measured VAE strengthens from first to last. If adaptation is compounding across cycles, it would at least be happening across all four adaptation durations (e.g., the proportional difference between the shortest and longest adaptation durations remains the same). However, if there’s sufficient data, looking at just the first adaptation-test cycle within the shortest adaptation condition could be an informative way to test your reference frame effects at the shortest possible time point.

Maximal (possibly overkill) suggestion: Test whether adaptation appears to compound across cycles, and separately analyze data from just the first cycle.

Intermediate suggestion: Test and report whether adaptation compounds across cycles, discuss whether this has any influence on your conclusions.

Minimal request: Discuss the possibility and how it might have influenced your results.

One final note in this section: While eye tracking would have been ideal, I do not think it is as critical as the previous reviewer did. Its primary value would probably have been in ensuring that participants were looking at the correct stimulus or fixation point, more so than assessing the precise accuracy of their fixation. Given the wide spacing between stimulus locations and the ample time participants have to correct initial saccade errors prior to stimulus presentation, fixation should have been accurate enough as long as participants followed directions.

Small things

A possible discussion point: Some studies have shown the VAE to be limited to the region of space where AV exposure occurred (e.g. Bruns & Röder, Psychological Res., 2019; earlier work cited in the intro of that paper). I’m curious what this means in the context of exposures where stimuli in one modality are spatially variable, while stimuli in the other modality are fixed in physical space (as is the case in your EC condition of Exp. 1). Does this affect how much the VAE generalizes during the test phase? I could take this or leave it in your discussion, though; it’s not as if you’re short for content.

L113) Please report participant demographics for the final pool after rejections, instead of / in addition to the pre-rejection demographics.

L228) So participants moved the visual marker and registered responses using peripheral vision? Just checking this.

L428) Figures 2a and 2b referenced in the text should be Figures 3a and 3b.

Reviewer #2: This paper aimed at investigating the contributions of eye- vs. head-centered reference frames for crossmodal spatial recalibration in the ventriloquism aftereffect (VAE) paradigm. To this end, three adaptation conditions were compared, one in which EC and HC reference frames were aligned, one in which the eyes (but not the head) followed the visual component but the actual AV spatial disparity was preserved, and one in which the auditory stimulus was always presented at ±20° relative to the head regardless of the (fixated) visual stimulus location. It was found that the VAE was reduced but still significant in the latter two conditions, and this effect did not interact with adaptation duration. A control experiment found a reduced but significant VAE also in a condition in which EC and HC were aligned but actual AV spatial disparities differed as in the incoherent reference frames of the main experiment.

The paper has already been through one round of revisions and while reading the manuscript I had the same thought as the previous reviewer: Why would auditory information stay in an exclusively head-centered reference frame and why/how would head-centered auditory information be integrated with eye-centered visual information without prior remapping? I think it should be made clearer early on (i.e. explicitly mentioned already in the introduction and methods) that the experiment design is based on the assumption that auditory information stays exclusively HC but can nevertheless be combined with EC visual information and that any conclusions about contributions of reference frames are only valid if this assumption is true. That way, the readers could better judge for themselves what to make of the results.

Having said that, however, the discussion of whether HC auditory coordinates could be directly integrated with EC visual coordinates somewhat reminded me of the literature on tactile remapping with hand crossing manipulations where anatomical and external coordinates are brought into conflict (e.g., Yamamoto & Kitazawa, 2001, Nat Neurosci). Some studies have suggested that visual-external coordinates may be integrated with anatomical (i.e., non-remapped) tactile coordinates if processing time is short (e.g., Azanon & Soto-Faraco, 2008, Curr Biol) or if the presumed tactile remapping process is disrupted via TMS (Bolognini & Maravita, 2007, Curr Biol; Renzi et al., 2013, J Cogn Neurosci). Thus, I think that these findings do support the authors’ line of argument, but I would leave it up to them whether they want to discuss these findings giving that the present study did not involve any tactile stimuli.

Moreover, I think the new control experiment at least partially addresses the issue and (in light of the Bruns et al. 2020 findings) the results of this new experiment are highly interesting in their own right. However, a statistical comparison of the new control condition is only carried out with the baseline condition in which both reference frames were aligned and I was wondering whether a statistical comparison with the other two conditions wouldn’t be informative as well? For example, one could have had the hypothesis that spatially incoherent in both RF leads to a stronger reduction of the VAE than incoherence in only one RF (which does not seem to be the case from visual inspection of the data).

There may be other implications of the results worthy being pointed out in the discussion. For example, many studies of the VAE did not control for eye movements during the exposure phase. The results of the head-coherent condition may suggest that these studies may nevertheless provide valid estimates of the VAE.

Another thought I had was whether the design with only very short (1 min) breaks between different adaptation duration blocks is really suitable to investigate effects of adaptation duration. Newer findings suggest that there will likely be carry-over effects between blocks (e.g., Bruns & Röder, 2019, JEP:HPP) and I wonder to what extent such carry-over effects might have obscured any effect of adaptation duration within a particular block?

Other than the points mentioned, I think this is a really nice and technically sound paper which is very well-written and uses well-justified statistical analyses, and I have no further comments of substance.

6. PLOS authors have the option to publish the peer review history of their article (what does this mean?). If published, this will include your full peer review and any attached files.

Reviewer #1: No

Reviewer #2: No

---

## [Author Response · Author response to Decision Letter 0]

19 Apr 2021

Please see attached "Response to Reviewers" document.

---

## [Decision Letter · Decision Letter 1]

4 May 2021

Multiple spatial reference frames underpin perceptual recalibration to audio-visual discrepancies

PONE-D-21-03641R1

Dear Dr. Watson,

We’re pleased to inform you that your manuscript has been judged scientifically suitable for publication and will be formally accepted for publication once it meets all outstanding technical requirements.

Kind regards,

Nicholas Seow Chiang Price, Ph.D.

Academic Editor

PLOS ONE

Additional Editor Comments (optional):

Reviewers' comments:

Reviewer's Responses to Questions

**Comments to the Author**

1. If the authors have adequately addressed your comments raised in a previous round of review and you feel that this manuscript is now acceptable for publication, you may indicate that here to bypass the “Comments to the Author” section, enter your conflict of interest statement in the “Confidential to Editor” section, and submit your "Accept" recommendation.

Reviewer #1: All comments have been addressed

Reviewer #2: All comments have been addressed

2. Is the manuscript technically sound, and do the data support the conclusions?

Reviewer #1: Yes

Reviewer #2: Yes

3. Has the statistical analysis been performed appropriately and rigorously? 

Reviewer #1: Yes

Reviewer #2: Yes

4. Have the authors made all data underlying the findings in their manuscript fully available?

Reviewer #1: Yes

Reviewer #2: Yes

5. Is the manuscript presented in an intelligible fashion and written in standard English?

Reviewer #1: Yes

Reviewer #2: Yes

6. Review Comments to the Author

Reviewer #1: (No Response)

Reviewer #2: (No Response)

7. PLOS authors have the option to publish the peer review history of their article (what does this mean?). If published, this will include your full peer review and any attached files.

Reviewer #1: **Yes: **Justin T Fleming

Reviewer #2: No

---

## [Editor Report · Acceptance letter]

6 May 2021

PONE-D-21-03641R1 

Multiple spatial reference frames underpin perceptual recalibration to audio-visual discrepancies 

Dear Dr. Watson:

I'm pleased to inform you that your manuscript has been deemed suitable for publication in PLOS ONE. Congratulations! Your manuscript is now with our production department. 

Kind regards, 

on behalf of

Dr. Nicholas Seow Chiang Price 

Academic Editor

PLOS ONE